# Characterization of Exterior Parts for 3D-Printed Humanoid Robot Arm with Various Patterns and Thicknesses

**DOI:** 10.3390/polym16070988

**Published:** 2024-04-04

**Authors:** Dikshita Chowdhury, Ye-Eun Park, Imjoo Jung, Sunhee Lee

**Affiliations:** 1Department of Fashion and Textiles, Dong-A University, Busan 49315, Republic of Korea; 2Department of Fashion Design, Dong-A University, Busan 49315, Republic of Korea

**Keywords:** robot exterior material, auxetic re-entrant structure, fused filament fabrication (FFF) 3D printing, thermoplastic polyurethane (TPU), mechanical property

## Abstract

Currently, metal is the most common exterior material used in robot development due to the need to protect the motor. However, as soft, wearable, and humanoid robots are gradually being developed, many robot parts need to be converted into artificial skin using flexible materials. In this study, in order to develop soft exterior parts for robots, we intended to manufacture exterior robot arm parts via fused filament fabrication (FFF) 3D printing according to various structural and thickness conditions and analyze their mechanical properties. The exterior parts of the robot arms were manufactured utilizing Shore 95 A TPU (eTPU, Esun, Shenzhen, China), which is renowned for its softness and exceptional shock absorption characteristics. The exterior robot arm parts were modeled in two parts, the forearm and upper arm, by applying solid (SL) and re-entrant (RE) structures and thicknesses of 1, 2, and 4 mm. The mechanical properties were analyzed through the use of three-point bending, tensile, and compression testing. All of the characterizations were analyzed using a universal testing machine (AGS-X, SHIMADZU, Kyoto, Japan). After testing the samples, it was confirmed that the RE structure was easily bendable towards the bending curve and required less stress. In terms of the tensile tests, the results were similar to the bending tests; to achieve the maximum point, less stress was required, and for the compression tests, the RE structure was able to withstand the load compared to the SL structure. Therefore, after analyzing all three thicknesses, it was confirmed that the RE structure with a 2 mm thickness had excellent characteristics in terms of bending, tensile, and compressive properties. Therefore, the re-entrant pattern with a 2 mm thickness is more suitable for manufacturing a 3D-printed humanoid robot arm.

## 1. Introduction

Traditionally, robot exteriors have predominantly been composed of rigid materials, limiting their physical interaction capabilities with humans. These exterior parts were designed to encase components such as motors, actuators, sensors, and associated electronic wiring to safeguard the internal mechanisms. Typically, metals or plastics were employed in the manufacturing process to ensure robust protection [1,2]. However, with the advancement of humanoid robotics, there has been a growing recognition of the necessity for softer and more flexible exteriors to facilitate improved interaction with humans. A soft robot exterior can be achieved through the careful selection of materials or by designing the structure accordingly [3]. An emerging and promising approach to crafting soft robot exteriors involves the utilization of auxetic structures in meta-materials. Auxetic structures exhibit a unique property wherein they can expand laterally in one or more perpendicular directions when subjected to axial extension. This distinctive behavior grants them desirable mechanical properties such as elasticity, shock absorption, and strain resilience. By incorporating auxetic meta-materials into the exterior composition, robots can benefit from enhanced protection against external impacts while still maintaining the required flexibility for enabling human interactions. This approach holds significant promise in advancing the field of robotics by enabling robots to operate safely and effectively in close proximity to humans [4,5].

Auxetic materials are structures that have a negative Poisson ratio [6] and are constructed in such a way that a gap exists between the structure of the sample, yielding properties of permeability, absorption, indentation resistance, shear resistance, toughness, and easy bending, resulting in synclasticity. In an auxetic structure, deformation starts from one cell, which is followed by the cells in the particular row [7,8,9]. The energy absorption properties of auxetic materials are excellent as the structure causes the material to swell when compressed, withstanding the load applied [10]. Due to their unique properties, they have found applications in industries like fashion, aerospace, automotive, sports, medical textiles, footwear, and architecture [9,11,12]. The structure of re-entrant (RE) auxetic materials means that the materials have a negative angle on all sides that directs inward like a bow tie, causing axial deformation when force is applied [13]. As the RE structure is not exactly symmetrical, it is affected differently in all directions when force is applied. The RE structure has more auxetic properties with smaller angles [14]. Three-dimensionally structured hexagonal RE has auxetic behaviors in multiple directions, and it was also found that the mechanical properties of a RE can also be controlled by the RE angle [15].

As auxetic materials tend to expand when force is applied, their structures are typically composed of softer, more flexible, and absorbent materials. Since the development of 3D printing (3DP), it has been most widely used to create such structures as it can create complex structures in less time with less wastage. The most commonly used 3DP method is fused filament fabrication (FFF). It is the simplest and easiest layering extruder that is widely used to build 3DP materials [16,17]. This technique enables precise control over both heating and nozzle size, allowing adjustments to be made according to specific requirements [18]. Thermoplastic polyurethane (TPU) is commonly used in FFF applications due to its good elasticity and thermoplastic processing characteristics. Furthermore, it belongs to the class of thermoplastic elastomers (TPE) relating to its mechanical flexibility [19]. It has both elastic and rigid thermoplastic characteristics. It can be easily deformed by heating and has a high elasticity [20,21,22]. In recent decades, researchers have studied 3DP auxetic structures because of their beneficial properties. It was found that RE structures can cause elastic bulking before collapsing, with it being noted that auxetic RE structures have the ability to consistently absorb mechanical, heat, and thermal energy when force is applied. Furthermore, their cells can fold and have proven to be more durable and force-resistant against energetic impacts, showing that these structures are more reliable [23,24]. Accordingly, TPU can be used to make auxetic structures, as previous studies have shown that TPU has good elongation properties, low crystallinity, absorbance, resistance, and toughness, giving it flexibility, resistance, and absorption properties [25,26].

Previously, a study was carried out on the motion control of 3D-printed (3DP) fingers with two types of samples: cap and RE. It was observed in the study that the RE samples were more suitable in terms of finger movements and grabbing as it was very easy to bend due to their synclasticity relating to their auxetic structure [27]. In another study of a 3DP wrist brace, it was observed that TPU has excellent shock absorption properties relating to its high negative Poisson ratio (NPR) and toughness [28]. TPU provides several benefits in terms of elasticity, shock absorption, high resistance, and easy structural customization.

This study aimed to develop a soft exterior cover for the original humanoid robot ALICE Ⅲ (AeiROBOT Inc., Gyeonggi-do, Korea) using FFF 3D printing with a Shore 95A TPU filament. This development is crucial as human–robot interactions are increasing, and this research has the goal of facilitating smoother interactions while also protecting the robot’s metal body parts from the surroundings [1,2]. Robotic arms are the parts that perform major movements via joint movements [29,30]. Accordingly, we designed the exterior material of ALICE Ⅲ’s robot arm and attempted to confirm the appropriate conditions. The exterior material of the robot arm was designed to be divided into two parts, the forearm and upper arm, for the movement of the elbow joint. To confirm the suitable conditions for the robot arm’s exterior material, two different patterns and three different thicknesses were applied to the 3DP sample. The mechanical properties were analyzed through bending, tensile, and compressive tests. Based on the research results, we intend to use this material to cover the arms of the humanoid robot.

## 2. Experimental Section

### 2.1. Materials

Figure 1 shows the original humanoid robot ALICE III (AeiROBOT Inc., Gyeonggi-do, Republic of Korea). The robot is 1360 mm in height, and the size of the forearm is 240 × 165 × 150 mm^3^, the upper arm is 275 × 190 × 150 mm^3^, and it weighs 25 kg and has 24 DOF (degree of freedom). To manufacture the 3DP humanoid robot arm with various patterns and thicknesses, a white thermoplastic polyurethane (TPU) filament (eTPU-95A, Esun Industrial Co. Ltd., Shenzhen, China) with a diameter of 1.75 mm, a density of 1.43 g/cm^3^, and a Shore hardness of 95A was utilized. Printing was conducted using a fused filament fabrication (FFF) technique with a 3D printer (Cubicon single plus, Cubicon Co. Ltd., Gyeonggi-do, Republic of Korea) equipped with a 0.4 mm diameter nozzle. The 3D model of the humanoid robot arm was saved in *.stl files. Slicing and printing were performed using the Cubicreator4 V4.4.0 slicing program (Cubicon Co. Ltd., Gyeonggi-do, Republic of Korea) under the following conditions: a nozzle temperature of 225 °C, a bed temperature of 65 °C, a printing speed of 60 mm/s, and infill pattern and density set to zigzag and 20%, respectively. The transformation of *.stl files into printable *.g-code files was executed within the slicing program.

### 2.2. Preparation of 3D-Printed Forearm and Upper Arm with Three Different Thicknesses

Table 1 shows the sample model and sample size. The 3DP samples were divided into two arm parts: forearm (FA) and upper arm (UA). The two parts were designed to be separated into FA and UA to facilitate easy movement of the elbow joint. Additionally, these were designed to fit the size of the robot arm, and button closure was applied to enable detachment. The size of the FA was modeled with a width of 130 mm and a length of 165 mm, and the UA was modeled with a width of 155 mm and a length of 190 mm using the Fusion 360 program (Autodesk Inc., San Francisco, CA, USA). In the case of thickness, thicknesses of 1, 2, and 4 mm were also modeled to confirm a more suitable thickness for the humanoid robot arm. In addition, two types of patterns, solid (SL) and re-entrant (RE), were applied. The sample code indicated the arm part (FA/UA), pattern (SL/RE), and thickness (1/2/4): FA_SL1, FA_SL2, FA_SL4, FA_RE1, FA_RE2, FA_RE4, UA_SL1, UA_SL2, UA_SL4, UA_RE1, UA_RE2, and UA_RE4.

### 2.3. Characterizations

#### 2.3.1. Analysis of Slicing of 3D-Printed Humanoid Robot Arm with Various Patterns and Thicknesses

The slicing program Cubicreator 4 v4.4.0 (Cubicon Inc., Gyeonggi-do, Republic of Korea) was used to slice all three (1, 2, and 4 mm) samples with different thicknesses and patterns (SL and RE). The properties of the two different patterns with three thicknesses were characterized using a slicing program, and the two types of actual printouts were compared and analyzed together. Firstly, in the slicing program, the 3D printing conditions were met and then printed through the preparation screen, and the infill shape inside the layer was solidified. It was confirmed with the model. Samples with a thickness of 4 mm consisted of a total of 32 layers, including 3 layers of the inner wall and 3 layers of outer wall, with the rest being infill layers. Samples with a 2 mm thickness had a total of 16 layers, consisting of inner and outer walls of 3 layers each and the rest being infill layers. Those samples with a 1 mm thickness only consisted of an inner and outer layer with a total of 8 layers as there was no space for infill layers. The thicknesses were set to 1, 2, and 4 mm. The actual sample characteristics were compared based on their mechanical properties. High-resolution images of the slicing analysis in the text are re-attached to Appendix A.

#### 2.3.2. Actual Printing Time and Weight of 3D-Printed Humanoid Robot Arm with Various Patterns and Thicknesses

The SL and RE of the FA and UA samples of 1, 2, and 4 mm thicknesses were compared by checking the actual time after printing. The actual time was checked to distinguish which samples have more efficiency in relation to the thickness and patterns [28]. The actual time was calculated by taking the ratio of the relative value of each thickness of both patterns. After 3D printing, the actual printing time was confirmed, as checked by the FFF 3D printer. The value of FA_SL, regardless of thickness and pattern, was considered the minimum value for calculating the ratio of actual time for both FA and UA, as it was observed that the FA_SL samples consistently took less time across various thicknesses during printing. The weight was measured with an analysis scale (PAG114, OHAUS, Parsippany, NJ, USA). Similar to the actual time, the actual weight was also checked to determine which pattern was lighter, facilitating smooth interactions. Therefore, the actual weight was utilized to compare the two patterns [31]. The average weight of three pieces of the same sample was taken after measuring. The weight was calculated by taking the ratio of the relative value of each thickness of both patterns. The value of FA_SL, irrespective of thickness and pattern, served as the baseline for calculating the ratio of the actual weight for both FA and UA. The decision was based on the observation that FA_SL consistently took less time to print across various thicknesses, making it the minimum value for the ratio of actual time.

#### 2.3.3. Mechanical Properties of 3D-Printed Humanoid Robot Arm with Various Patterns and Thicknesses

##### Bending Properties

In terms of bending properties, the test was conducted according to the methods used in previous studies [10,12]. Figure 2 shows the universal mechanical testing machine (AGS-X, Shimadzu Co. Ltd., Kyoto, Japan) used to carry out the bending test to determine the bending characteristics of FA and UA samples with various thicknesses. Three-point bending properties were measured based on the KS M ISO 14125 [32]. The gauge size was 75 mm × 25 mm, and the bending speed was 2 mm/min with a load of 5 kN. The sample was pressed to strain until the highest bending point. A stress–strain (S–S) curve was obtained. Based on the S–S curve, bending initial modulus, bending maximum stress, bending maximum strain, and bending toughness were measured.

##### Tensile Properties

For the tensile properties, the test was conducted according to the methods used in previous studies [10,12]. Figure 3 shows the universal mechanical testing machine used for the tensile tests to determine the tensile characteristics of the FA and UA samples with various thicknesses. The tensile properties were measured based on the KS K 0520 [33]. The gauge length was 75 mm × 25 mm, and the tensile speed was 25 mm/min with a load of 5 kN. The sample was stretched to elongation until the sample reached its maximum strain and broke. A S–S curve was obtained. Based on the S–S curve, tensile initial modulus, tensile maximum stress, tensile maximum strain, and tensile toughness were measured.

##### Compressive Property

For the compressive properties, the test was conducted according to the methods used in previous studies [10,12]. Figure 4 shows the universal mechanical testing machine used for the compressive test to determine the compressive characteristics of the FA and UA samples with various thicknesses. The compressive properties were measured based on KS M ISO 604 [34]. The gauge diameter was 40 mm, and the compression speed was 10 mm/min with a load of 5 kN. The sample was compressed until it reached its maximum strain. A S–S curve was obtained. Based on the S–S curve, compressive initial modulus, compressive maximum stress, compressive maximum strain, and compressive toughness were measured.

## 3. Results and Discussion

### 3.1. Analysis of Sliced Images for Modeling of 3D-Printed Humanoid Robot Arm with Various Patterns and Thicknesses

Table 2 displays the top and front views of the sliced images of the 3DP models, while Table 3 showcases images of the 3DP samples. These images reveal significant differences in the infill patterns across various thicknesses and between the two distinct patterns used, SL and RE. From the table, we can see the differences in the infill between all three thicknesses and the two patterns. It can be observed that SL1 only consists of inner and outer walls comprising five layers and no infill. In contrast, SL2 features infill with a total of 10 layers, including the inner and outer walls. Furthermore, SL4 exhibits more prominent infill, with inner and outer walls spanning 20 layers, demonstrating the highest thickness. For precise analysis of the differences between the two patterns, we suggest referring to Appendix A for high-resolution images. In the case of the RE pattern, from Table 2, it can be observed that the sample was predominantly covered by the inner and outer layers relating to its structure, leaving very little space for infill compared to the SL pattern, despite the slicing conditions being the same for both patterns. Specifically, RE1 consists of a total of five layers, comprising only inner and outer walls. RE2 consists of 11 layers, including infill, while RE4 exhibits the most infill, with 20 layers of TPU filament. For a detailed examination of the distinctions between the two patterns, it is advised to consult Appendix A for access to high-resolution images. This analysis highlights the distinct infill characteristics between SL and RE patterns, providing insights into their structural integrity and potential implications for mechanical properties such as the stiffness, flexibility, and durability of the 3DP samples [35].

### 3.2. Actual Printing Time and Weight of 3D-Printed Humanoid Robot Arm with Various Patterns and Thicknesses

Figure 5 shows the actual printing time and ratio of the samples for the FA and UA samples with SL and RE patterns. For FA_SL, it was confirmed that the printing time for FA_SL1 was 2 h 19 m 30 s; for FA_SL2, it was 3 h 1 m 34 s; and for FA_SL4, it was 3 h 36 m 50 s. For FA_RE, it was confirmed that the printing time for FA_RE1 was 2 h 59 m 49 s; for FA_RE2, it was 4 h 54 m 19 s; and for FA_RE4, it was 8 h 2 m 28 s. The actual time was determined by calculating the ratio of the relative value of each thickness. As the value of FA_SL, regardless of thickness, was considered the minimum value for calculating the ratio of actual time, it was confirmed to be 1.00 for FA_SL1, 1.00 for FA_SL2, 1.00 for FA_SL4, 1.28 for FA_RE1, 1.62 for FA_RE2, and 2.22 for FA_RE4. For UA_SL, it was confirmed that the printing time for UA_SL1 was 3 h 4 m 4 s; for UA_SL2, it was 3h 58 m 4 s; and for UA_SL4, it was 4 h 39 m 30 s. For FA_RE, it was confirmed that the printing time for UA_RE1 was 3 h 54 m 52 s; for UA_RE2, it was 6 h 19 m 29 s; and for UA_RE4, it was 10 h 19 m 25 s. As the actual printing time ratio was calculated, it was confirmed to be 1.32 for UA_SL1, 1.31 for UA_SL2, 1.28 for UA_SL4, 1.67 for UA_RE1, 2.09 for UA_RE2, and 2.85 for UA_RE4. The key observation is that the RE pattern consistently required more time for printing compared to the SL pattern, regardless of the thickness or complexity of the pattern. Additionally, the printing times varied based on both thickness and pattern, with thicker and more intricate designs generally requiring more time. These results are in line with expectations in additive manufacturing, where thicker layers and complex shapes naturally lead to longer printing times due to the deposition processes involved [31,35].

Figure 6 shows the weight and ratio of the samples for the FA and UA samples with SL and RE patterns. It was confirmed that the weight and ratio for FA_SL1 was 18.6 ± 0.2 g; for FA_SL2, it was 24.9 ± 1.5 g; and for FA_SL4, it was 34.2 ± 1.2 g. In the case of FA_RE, it was confirmed that the weight and ratio for FA_RE1 was 13.8 ± 2.2 g; for FA_RE2, it was 22.6 ± 0.1 g; and for FA_RE4, it was 40.2 ± 0.6 g. For UA_SL, it was confirmed that the weight and ratio for UA_SL1 was 24.2 ± 1.1 g; for UA_SL2, it was 34.1 ± 0.9 g; and for UA_SL4, it was 45.3 ± 0.6 g. In UA_RE, it was confirmed that the weight and ratio for UA_RE1 was 16.1 ± 0.1 g; for UA_RE2, it was 28.8 ± 0.3 g; and for UA_RE4, it was 53.9 ± 1.0 g. The actual weight of FA was determined by calculating the ratio of the relative value of each thickness. As the value of FA_SL, irrespective of thickness, served as the baseline for calculating the ratio, it was confirmed to be 1.00 for FA_SL1, 1.00 for FA_SL2, 1.00 for FA_SL4, 0.74 for FA_RE1, 0.90 for FA_RE2, and 1.17 for FA_RE4. The actual printing weight ratio of UA was calculated, and it was confirmed to be 1.30 for UA_SL1, 1.36 for UA_SL2, 1.32 for UA_SL4, 0.86 for UA_RE1, 1.15 for UA_RE2, and 1.57 for UA_RE4. The RE4 thickness tended to produce heavier samples, while the FA_RE1, FA_RE2, and UA_RE1 patterns were lighter than their counterparts in the FA_SL pattern. The differences in weight are due to the fact that thicker layers and complex shapes require longer printing times. This is related to the deposition process, where material is added layer by layer, which takes more time when dealing with thicker layers and intricate shapes [31,35].

### 3.3. Mechanical Properties of 3D-Printed Humanoid Robot Arm with Various Patterns and Thicknesses

#### 3.3.1. Bending Properties

Figure 7 shows the S–S curve of the bending test. Figure 8 shows the bending properties of the FA and UA samples with SL and RE patterns. From Figure 7a,b show the bending stress and strain of FA_SL and FA_RE, and it can be seen that the SL pattern normally withstood more stress and strain than the RE pattern except for 1 mm thickness. In regard to thickness, FA_RE1 has the highest stress of 0.45 ± 0.20 MPa and strain of 1.46 ± 1.16%. In Figure 7c,d show that the bending stress (MPa) of the SL pattern is greater than that of the RE pattern except for 1 mm thickness. In regard to any thickness and pattern, UA_SL2 has the highest stress of 0.62 ± 0.01 MPa. It was observed that the initial modulus for FA_SL1 was 3.89 ± 0.32 MPa; for FA_SL2, it was 3.91 ± 0.96 MPa; for FA_SL4, it was 2.15 ± 0.15 MPa; for FA_RE1, it was 2.66 ± 4.60 Mpa; for FA_RE2, it was 2.97 ± 0.10 MPa; and for FA_RE4, it was 1.64 ± 0.21 MPa. The result suggests that the initial modulus values for the SL pattern are higher than those for the RE pattern. Therefore, it can be concluded that the SL pattern generally exhibits higher stiffness or rigidity compared to the RE pattern. For UA_SL1, it was 10.56 ± 3.55 MPa; for UA_SL2, it was 14.22 ± 0.48 MPa; for UA_SL4, it was 2.42 ± 0.49 MPa; for UA_RE1, it was 2.74 ± 2.4 MPa; for UA_RE2, it was 3.74 ± 0.84 MPa; and for UA_RE4, it was 0.98 ± 0.23 MPa. Similar to FA in UA as well, the data indicate that the RE pattern generally exhibited lower initial modulus values compared to the SL pattern, implying greater flexibility and elasticity in the RE pattern. From stress_max_, it was seen that the value for FA_SL1 was 0.28 ± 0.03 MPa; for FA_SL2, it was 0.23 ± 0.01 MPa; for FA_SL4, it was 0.21 ± 0.01 MPa; for FA_RE1, it was 0.43 ± 0.20 MPa; for FA_RE2, it was 0.13 ± 0.10 MPa; and for FA_RE4, it was 0.17 ± 0.01 MPa. It was seen that the maximum stress values for the SL pattern were generally higher than those for the RE pattern. This observation suggests that the SL pattern tends to make the material stiffer, as it withstands higher stress levels compared to the RE pattern. For UA, it was observed that the value for UA_SL1 was 0.32 ± 0.03 MPa; for UA_SL2, it was 0.62 ± 0.01 MPa; for UA_SL4, it was 0.21 ± 0.02 MPa; for UA_RE1, it was 0.34 ± 0.11 MPa; for UA_RE2, it was 0.21 ± 0.00 MPa; and for UA_RE4, it was 0.12 ± 0.01 MPa. Similar to the initial modulus, the stress_max_ data indicate that the SL pattern generally results in higher stress values compared to the RE pattern, implying greater stiffness in the samples of SL pattern. The strain_max_ was observed in FA to be FA_SL1 1.92 ± 1.45%, FA_SL2 5.02 ± 0.36%, FA_SL4 10.17 ± 1.00%, FA_RE1 1.46 ± 1.16%, FA_RE2 3.30 ± 2.86%, FA_RE4 11.25 ± 0.51%. It was seen that the RE pattern typically results in lower strain_max_ compared to the SL pattern, except for FA_RE4, where the characteristics of its thickness and complex pattern led to higher strain_max_. In terms of UA, for UA_SL1, it was 1.83 ± 0.23%; for UA_SL2, it was 5.05 ± 0.14%; for UA_SL4, it was 11.15 ± 1.15%; for UA_RE1, it was 1.64 ± 1.02%; for UA_RE2, it was 5.31 ± 0.68%; and for UA_RE4, it was 11.04 ± 1.87%. Again, the RE pattern displayed lower strain_max_ values compared to the SL pattern, indicating higher flexibility. For toughness, FA_SL1 was 0.00 ± 0.00 J, FA_SL2 was 0.00 ± 0.00 J, FA_SL4 was 0.02 ± 0.00 J, FA_RE1 was 0.00 ± 0.00 J, FA_RE2 was 0.00 ± 0.00 J, and FA_RE4 was 0.01 ± 0.00 J. Both SL and RE patterns demonstrate low toughness, but the RE pattern shows slightly better energy absorption capabilities compared to the SL pattern. In UA, UA_SL1 was 0.00 ± 0.00 J, UA_SL2 was 0.01 ± 0.00 J, UA_SL4 was 0.02 ± 0.00 J, UA_RE1 was 0.00 ± 0.00 J, UA_RE2 was 0.00 ± 0.00 J, and UA_RE4 was 0.01 ± 0.00 J. Similar to FA in UA as well, both SL and RE patterns showed relatively low toughness values, with the RE pattern having slightly lower values than the SL pattern.

In Figure 8, it can be observed that regardless of thickness, the UA_SL2 pattern displays the highest initial modulus and stress at maximum load, indicating stiffness. Additionally, it can be noted that the RE patterns generally exhibit lower stress and strain at maximum load except for 1 mm thickness compared to the SL patterns. Despite this, the FA_RE4 pattern has the highest strain at maximum for the forearm sample. SL patterns demand more strain than RE patterns, with UA_SL4 showing the highest strain for the upper arm sample. In terms of toughness, FA_SL4 exhibits the highest toughness across patterns and thicknesses. The RE pattern sample had similar elongation to the SL pattern, while having lower strength. Thus, it was confirmed to be softer, tougher, and more elastic when bending.

Based on the results of bending property analysis, it was observed that the SL pattern exhibited greater stiffness, which was evidenced by larger initial modulus, maximum stress, maximum strain, and toughness. Conversely, the RE pattern showed lower values for these properties, indicating greater flexibility. The analysis focused on bending properties, comparing the SL pattern with the RE pattern. Regardless of thickness and pattern variation, the RE pattern with a 1 mm thickness demonstrated the highest flexibility and ease of bending. When subjected to load, the SL pattern resisted bending by shrinking perpendicularly, while the RE pattern formed a curve due to its auxetic (expandable) nature [36,37,38]. This characteristic made the RE pattern more flexible in the bending direction. However, as thickness increased, the RE pattern became stiffer and less bendable. The 4 mm thickness of the RE pattern was especially rigid compared to the 1 mm and 2 mm thicknesses, attributed to its thicker morphology. The RE pattern with a 1 mm thickness was very flexible but not suitable for the humanoid robot because it was too thin. Thin materials like this may bend easily but lack the strength and durability needed to support the robot’s functions. Therefore, despite its flexibility, it was not strong or durable enough for the robot’s needs [39]. Therefore, the 2 mm RE pattern was considered more suitable for the robot arm exterior, as it provided a balance of bending characteristics and elongation properties. Overall, the analysis highlights the trade-offs between stiffness and flexibility in different patterns and thicknesses.

#### 3.3.2. Tensile Properties

Figure 9 shows the S–S curve of the tensile test. Figure 10 shows the tensile properties of the FA and UA samples with SL and RE patterns. In Figure 9a,b > indicate that FA_SL1 has the highest breaking point, with a stress of 10.50 ± 0.54 MPa along with a strain of 1051.66 ± 20.97%. In Figure 9c,d indicate that the tensile max stress (MPa) appeared with the highest breaking point, with a stress of 16.62 ± 0.09 MPa and the highest strain of 990.35 ± 9.02% for UA_SL1.

The initial modulus, maximum stress (stress_max_), maximum strain (strain_max_), and toughness were measured for each pattern and thickness. It was seen that the initial modulus for FA was as follows: for FA_SL1, it was 9.56 ± 0.45 MPa; for FA_SL2, it was 4.16 ± 0.21 MPa; for FA_SL4, it was 1.73 ± 0.05 MPa; for FA_RE1, it was 2.47 ± 0.24 MPa; for FA_RE2, it was 0.81 ± 0.05 MPa; and for FA_RE4, it was 0.98 ± 0.02 MPa. Overall analysis suggests that the SL pattern shows higher initial modulus values with greater variability, while the RE pattern demonstrates lower but more consistent initial modulus values. In UA, it was found to be as follows: for UA_SL1, it was 20.77 ± 1.49 MPa; for UA_SL2, it was 12.08 ± 0.90 MPa; for UA_SL4, it was 6.58 ± 0.17 MPa; for UA_RE1, it was 1.94 ± 0.10 MPa; for UA_RE2, it was 1.57 ± 0.03 MPa; and for UA_RE4, it was 1.28 ± 0.05 MPa. It was seen from the data that, similar to FA, the SL pattern exhibited higher initial modulus values than the RE pattern across different thicknesses, implying flexibility in the RE pattern. From stress_max_, it was seen that for FA_SL1, it was 10.50 ± 0.54 MPa; for FA_SL2, it was 6.28 ± 0.09 MPa; for FA_SL4, it was 3.81 ± 0.18 MPa; for FA_RE1, it was 2.62 ± 0.21 MPa; for FA_RE2, it was 1.79 ± 0.09 MPa; and for FA_RE4, it was 1.46 ± 0.10 MPa. It was observed that the SL pattern requires greater forces for deformation with more variability in terms of measurements, while the RE pattern requires lower forces with relatively consistent measurements. In the case of UA, it was observed that for UA_SL1, it was 16.62 ± 0.09 MPa; for UA_SL2, it was 8.46 ± 0.52 MPa; for UA_SL4, it was 4.74 ± 0.18 MPa; for UA_RE1, it was 3.33 ± 0.35 MPa; for UA_RE2, it was 2.09 ± 0.19 MPa; and for UA_RE4, it was 1.92 ± 0.13 MPa. The results of stress_max_ also tended to be higher for the SL pattern compared to the RE pattern, similar to FA. Strain_max_ was observed in FA_SL1 to be 1051.66 ± 20.97%, and for FA_SL2, it was 1030.47 ± 83.32 MPa; for FA_SL4, it was 1019.79 ± 61.24%; for FA_RE1, it was 498.47 ± 76.83%; for FA_RE2, it was 497.89 ± 32.00%; and for FA_RE4, it was 457.73 ± 57.75%. The SL pattern tended to yield higher maximum strain values, whereas the RE pattern generally produced lower maximum strain values. As for UA, it was observed that for UA_SL1, it was 990.35 ± 9.02%; for UA_SL2, it was 866.01 ± 68.40%; for UA_SL4, it was 751.60 ± 34.91%; for UA_RE1, it was 580.39 ± 91.58%; for UA_RE2, it was 454.72 ± 114.87%; and for UA_RE4, it was 744.86 ± 108.16%. It was seen that the SL pattern generally required more strain than the RE pattern regardless of thicknesses. In terms of toughness, FA_SL1 was 47.65 ± 2.32 J, FA_SL2 was 56.99 ± 4.69 J, FA_SL4 was 72.59 ± 4.43 J, FA_RE1 was 5.05 ± 1.21 J, FA_RE2 was 7.25 ± 0.56 J, and FA_RE4 was 11.88 ± 2.25 J. The SL pattern demonstrates higher toughness values, suggesting better energy absorption capabilities compared to the RE pattern, although the RE pattern shows some improvement in toughness with increased thickness. For UA, the value for UA_SL1 was 64.33 ± 1.98 J; for UA_SL2, it was 62.69 ± 8.09 J; for UA_SL4, it was 63.24 ± 4.64 J; for UA_RE1, it was 8.02 ± 2.15 J; for UA_RE2, it was 7.72 ± 2.59 J; and for UA_RE4, it was 25.57 ± 7.57 J. Similar to the initial modulus, stress, and strain, the SL pattern tended to have higher toughness values than the RE pattern.

In Figure 10, it can be seen that both FA and UA showed that the SL pattern was stiffer than the RE pattern. UA_SL1 was the stiffest regardless of thickness, with an initial modulus of 20.77 ± 1.49 MPa. The SL pattern also required higher stress and strain at maximum load compared to the RE pattern. FA_SL1 had the highest strain, while FA_SL4 showed the highest toughness. It was observed that the SL pattern generally exhibited higher initial modulus and stress_max_ values compared to the RE pattern, indicating greater stiffness. Conversely, the RE pattern showed lower but consistent strain_max_ values, suggesting greater flexibility and elongation. This trend was consistent across different thicknesses. Overall, the data suggest that the RE pattern exhibits characteristics of soft compared to the SL pattern, which was observed to be stiffer, with a higher breaking point at high stress levels and samples failing to break at a certain point.

The tensile property analysis revealed that the SL pattern required more force to break the sample than the RE pattern due to its larger initial modulus, maximum stress, maximum strain, and toughness. Notably, the SL pattern with a 4 mm thickness exhibited the highest elongation before reaching the maximum stress point, regardless of thickness or pattern. Furthermore, the SL pattern consistently demonstrated more elongation, with the sample failing to break at a certain point. The stiffness of the SL pattern was maintained until reaching maximum stress, even as the load continued. In contrast, the RE pattern exhibited greater permeability compared to the SL pattern, expanding as it stretched and breaking at the maximum breaking point, indicating its shock absorption capabilities. The RE pattern displayed auxetic behavior, characterized by the lateral expansion of the sample and subsequent collapse of the cells in the affected row [8,38,40]. However, the 1 mm thickness of the RE pattern resulted in stiffness with more elongation, while the 4 mm thickness was too stiff but prone to easy breakage, suggesting poor shock-absorbing properties. Therefore, it was concluded that the RE pattern with a 2 mm thickness was the most suitable option due to its elongation characteristics and good shock absorption capabilities.

#### 3.3.3. Compressive Property

Figure 11 shows the compressive S–S curves. Figure 12 shows the compressive properties of the FA and UA samples with SL and RE patterns.

It was observed that the compressive initial modulus for FA_SL1 was 718.37 ± 18.86 MPa; for FA_SL2, it was 356.27 ± 146.21 MPa; for FA_SL4, it was 134.70 ± 1.96 MPa; for FA_RE1, 70.48 ± 10.22 MPa; for FA_RE2, it was 115.34 ± 1.67 MPa; and for FA_RE4, it was 118.40 ± 35.13 MPa. In UA, it was found that for UA_SL1, the value was 729.14 ± 30.16 MPa; for UA_SL2, it was 408.00 ± 33.66 MPa; for UA_SL4, it was 164.49 ± 1.66 MPa; for UA_RE1, it was 54.62 ± 7.55 MPa; for UA_RE2, it was 3.71 ± 0.41 MPa; and for UA_RE4, it was 747.32 ± 11.79 MPa. In the case of the SL pattern, the value decreases as the thickness increases, whereas the value of the RE pattern tends to increase as the thickness increases. For compressive stress at 50%, it was seen that the value for FA_SL1 was 340.85 ± 7.24 MPa; for FA_SL2, it was 298.48 ± 26.38 MPa; for FA_SL4, it was 69.85 ± 0.56 MPa; for FA_RE1, it was 127.29 ± 5.65 MPa; for FA_RE2, it was 185.78 ± 1.29 MPa; and for FA_RE4, it was 271.30 ± 2.84 MPa. As for UA, it was observed that the value for UA_SL1 was 335.47 ± 6.36 MPa; for UA_SL2, it was 309.02 ± 5.43 MPa; for UA_SL4, it was 108.00 ± 3.08 MPa; for UA_RE1, it was 146.87 ± 5.61 MPa; for UA_RE2, it was 220.62 ± 3.40 MPa; and for UA_RE4, it was 231.13 ± 23.48 MPa. The trend observed in the SL pattern indicates a decrease in value with increasing thickness, mirroring the behavior of the initial modulus. Conversely, the RE pattern exhibits a tendency for value augmentation with thicker specimens. Additionally, the SL pattern was generally superior to the RE pattern in strength at 50%. In the case of 4 mm thickness, the RE pattern was found to be larger. The compressive strain_max_ value observed in FA_SL1 was 58.58 ± 0.96%; for FA_SL2, it was 59.57 ± 2.78%; for FA_SL4, it was 78.93 ± 0.26%; for FA_RE1, it was 93.45 ± 0.69%; for FA_RE2, it was 92.25 ± 0.14%; and for FA_RE4, it was 66.54 ± 0.61%. For UA, it was seen that the value for UA_SL1 was 59.44 ± 1.13%; for UA_SL2, it was 59.56 ± 0.84%; for UA_SL4, it was 71.23 ± 0.19%; for UA_RE1, it was 97.02 ± 0.18%; for UA_RE2, it was 78.34 ± 0.59%; and for UA_RE4, it was 68.61 ± 1.68%. In the case of compressive strain max, the value of the SL pattern increased as the thickness increased, and the value of the RE pattern decreased. In addition, the SL value was generally measured in a small range, confirming that it was harder. The compressive toughness value for FA_SL1 was 1.46 ± 0.02 J; for FA_SL2, it was 2.41 ± 0.09 J; for FA_SL4, it was 3.64 ± 0.02 J; for FA_RE1, it was 1.68 ± 0.02 J; for FA_RE2, it was 3.40 ± 0.02 J; and for FA_RE4, it was 5.88 ± 0.12 J.

In comparison to the SL pattern, the RE pattern exhibited higher toughness values, indicating the material’s ability to absorb more energy before fracturing is enhanced under RE conditions. However, the SL pattern also showed respectable toughness values, albeit generally lower than those observed under the RE pattern. In UA, the value for UA_SL1 was 1.49 ± 0.03 J; for UA_SL2, it was 2.54 ± 0.10 J; for UA_SL4, it was 3.45 ± 0.02 J; for UA_RE1, it was 1.91 ± 0.04 J; for UA_RE2, it was 3.01 ± 0.05 J; and for UA_RE4, it was 6.16 ± 0.14 J. Like FA, the RE pattern generally shows higher toughness values compared to the SL pattern.

From Figure 12, it is confirmed that the compressive initial modulus and stress at 50% showed similar tendency. As the thickness increased, the SL pattern value decreased and RE pattern increased. Additionally, the SL pattern was found to have a higher compressive stress than the RE pattern. In the case of strain_max_, the value of the SL pattern increased as the thickness increased, and the value of the RE pattern decreased. In the case of toughness, the value increased as the thickness increased for both SL and RE patterns. Additionally, the toughness of the RE pattern was confirmed to be superior. Therefore, the SL pattern was rigid. The data suggest that the RE pattern offers greater flexibility, elongation, toughness, and elasticity characteristics under compression compared to the SL pattern.

After analyzing the compressive properties, it was revealed that the RE pattern can undergo more compression compared to the SL pattern, suggesting that it has greater shock absorption capabilities. This means that when a force is applied to the RE pattern, it can compress more before reaching its breaking point, absorbing more energy in the process. The RE pattern displayed auxetic behavior, which means that it showed enhanced resistance to deformation compared to the SL pattern. As a result, when compressed, the SL pattern expanded laterally, causing the material to shrink. In contrast, the RE pattern resisted deformation and maintained its structure better under compression. The RE pattern’s ability to withstand compression can be attributed to its lower infill and predominantly inner and outer wall structure. This means that instead of deforming or collapsing under pressure, the material expanded, further enhancing its shock absorption capabilities [10,41]. Regarding thickness, in the case of the SL pattern, the strength decreased as the thickness increased. In the case of RE patterns, the opposite trend appeared. It was confirmed that the SL pattern had the same effect as foam material as the number of infill parts between top and bottom layers increased as the thickness increased. In the case of the RE pattern, even a 1 mm sample is believed to exhibit auxetic performance and be able to withstand the load. Based on these findings, it was concluded that the RE pattern with a 2 mm thickness is more suitable for use in humanoid robot arm exteriors due to its compressive characteristics. Overall, the analysis highlights the superior compressive properties of the RE pattern compared to the SL pattern, emphasizing its suitability for use in the exteriors of humanoid robot arms.

## 4. Conclusions

In this study, we aimed to develop a soft exterior material for humanoid robot arms using fused filament fabrication (FFF) 3D printing. Using a Shore 95 A TPU filament, the soft exteriors were developed with two different patterns and three different thicknesses, which were printed and then analyzed in terms of their mechanical properties.

The results are summarized as follows: the analysis of the slicing patterns of both the SL and RE patterns revealed notable differences. The RE pattern showed that the sample was mostly covered with inner and outer layers due to its structure, with very little space for infill compared to the SL pattern, which primarily consisted of infill except for the 1 mm sample due to its thinness. Despite the same slicing conditions for both patterns, the RE pattern consistently took more time to complete than the SL pattern. In terms of actual weight, the RE sample with a thickness of 4 mm had the highest weight among all of the samples.

In the bending property analysis, it was observed that when the load was applied to the SL pattern, it began to shrink perpendicularly, making it stiffer and more resistant to bending toward the curve. In contrast, the RE pattern started to form a curve due to its auxetic nature, making it more flexible and easily bendable toward the curve. The re-entrant pattern maintained its flexibility throughout, meaning it could withstand bending forces without breaking or excessive deformation. Similarly, in the tensile property analysis, the solid pattern required more stress (MPa) compared to the re-entrant pattern to achieve the maximum point in tensile strength. It could withstand high levels of tensile stress without reaching its breaking point. The RE pattern exhibited auxetic behavior, with the lateral expansion of the sample being noted, where deformation occurred from one cell, followed by the collapse of all cells in the particular row of the sample. The SL pattern maintained its stiffness until reaching maximum stress as the load continued. In the compressive property, it was confirmed that the RE pattern demonstrated enhanced resistance compared to the SL pattern, causing the SL pattern to expand laterally, resulting in material shrinkage. It could withstand compression forces effectively without collapsing or significant deformation. Furthermore, among both patterns and all three thicknesses, the RE pattern samples with 1 mm and 2 mm thicknesses exhibited more desirable characteristics for use in exterior parts. However, 1 mm thickness is the thinnest option, with anything thinner posing challenges in terms of ensuring smooth interactions and protecting the robot’s exterior parts from the surroundings [39].

According to the results of this study, using a 2 mm thick RE pattern is ideal for making a 3DP humanoid robot arm. The RE structure’s great bending, tensile, and compressive qualities make it a perfect choice for ensuring the strength and performance of a robot arm, highlighting its excellent mechanical properties and suitability for the intended application.

## Figures and Tables

**Figure 1 polymers-16-00988-f001:**
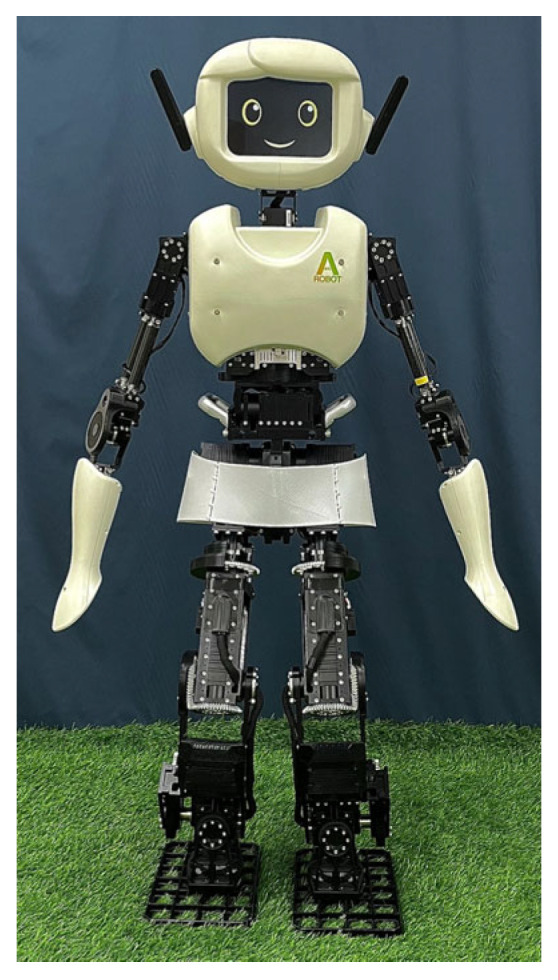
Humanoid robot ALICE III (AeiROBOT Inc., Korea).

**Figure 2 polymers-16-00988-f002:**
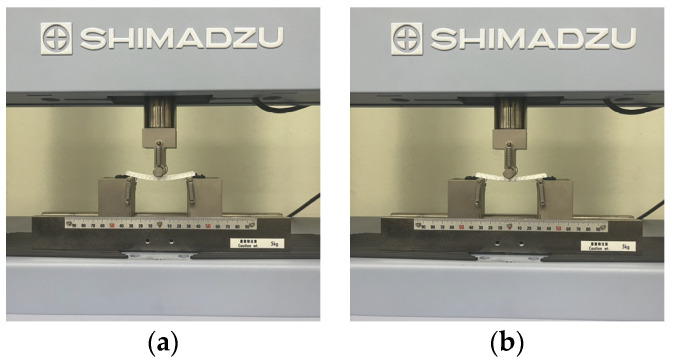
Mechanical property characterization process for 3D-printed forearm and upper arm for bending test: (**a**) solid pattern; (**b**) re-entrant pattern.

**Figure 3 polymers-16-00988-f003:**
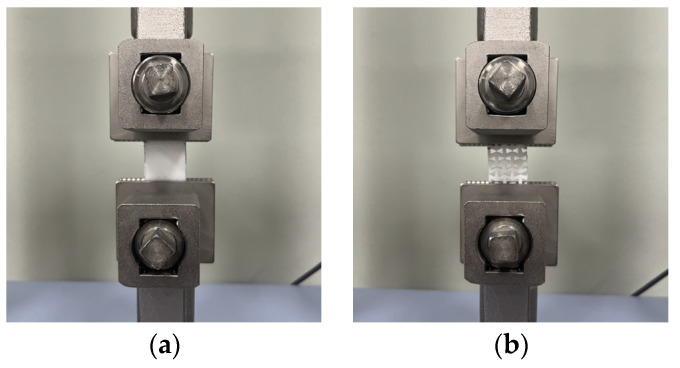
Mechanical property characterization process for 3D-printed forearm and upper arm for tensile test: (**a**) solid pattern; (**b**) re-entrant pattern.

**Figure 4 polymers-16-00988-f004:**
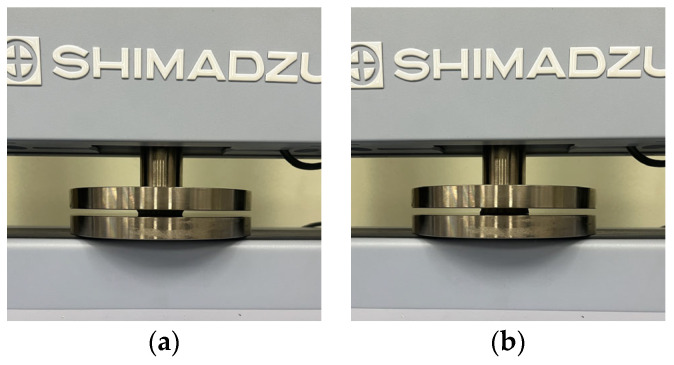
Mechanical property characterization process for 3D-printed forearm and upper arm for compressive test: (**a**) solid pattern; (**b**) re-entrant pattern.

**Figure 5 polymers-16-00988-f005:**
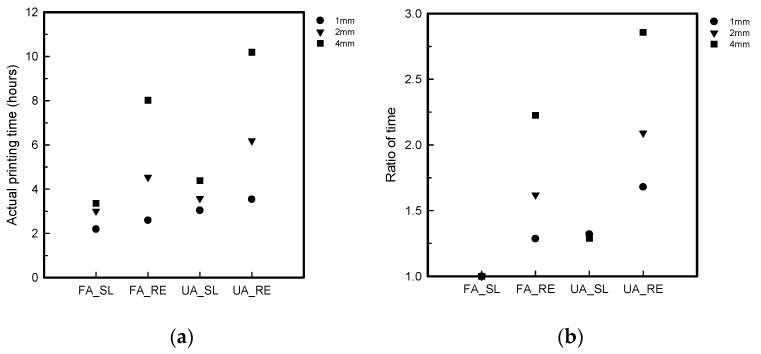
Analytical representation of 3D-printed forearm and upper arm samples with three different thicknesses: (**a**) actual printing time; (**b**) ratio of time.

**Figure 6 polymers-16-00988-f006:**
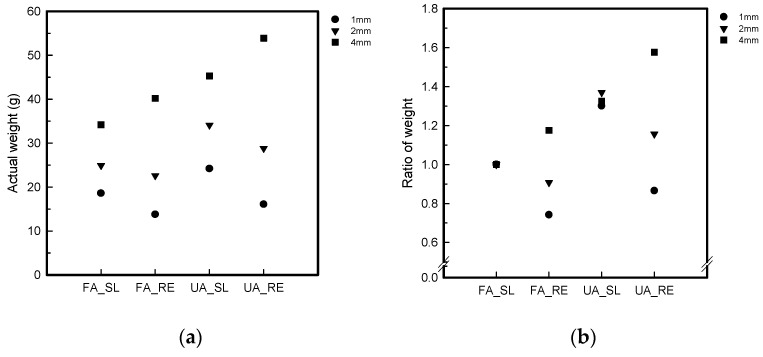
Analytical presentation of 3D-printed forearm and upper arm with three different thicknesses: (**a**) actual weight; (**b**) ratio of weight.

**Figure 7 polymers-16-00988-f007:**
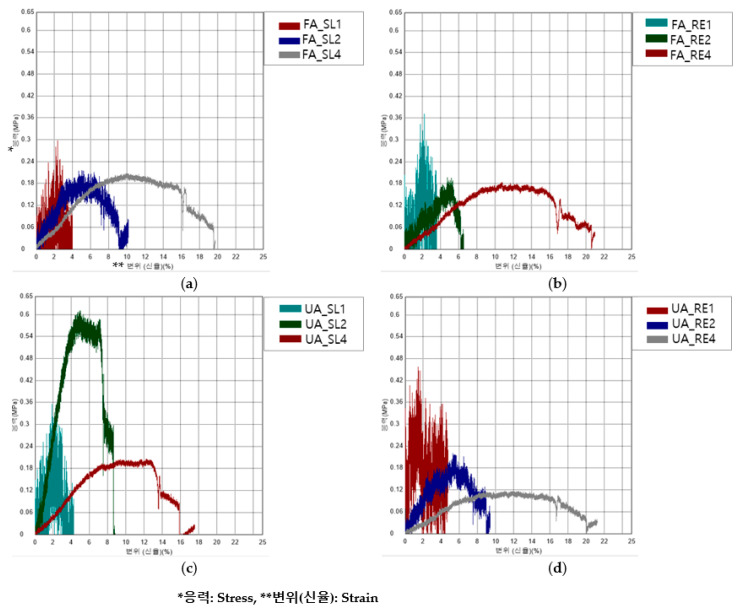
S–S curves of bending property of 3D-printed forearm and upper arm samples with three different thicknesses; (**a**) FA_SL, (**b**) FA_RE, (**c**) UA_SL, and (**d**) UA_RE.

**Figure 8 polymers-16-00988-f008:**
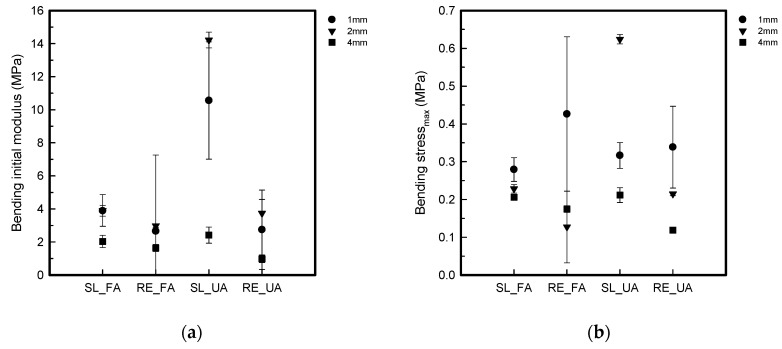
Bending properties of 3D-printed forearm and upper arm samples with three different thicknesses: (**a**) bending initial modulus; (**b**) bending stress_max_; (**c**) bending strain_max,_; (**d**) bending toughness.

**Figure 9 polymers-16-00988-f009:**
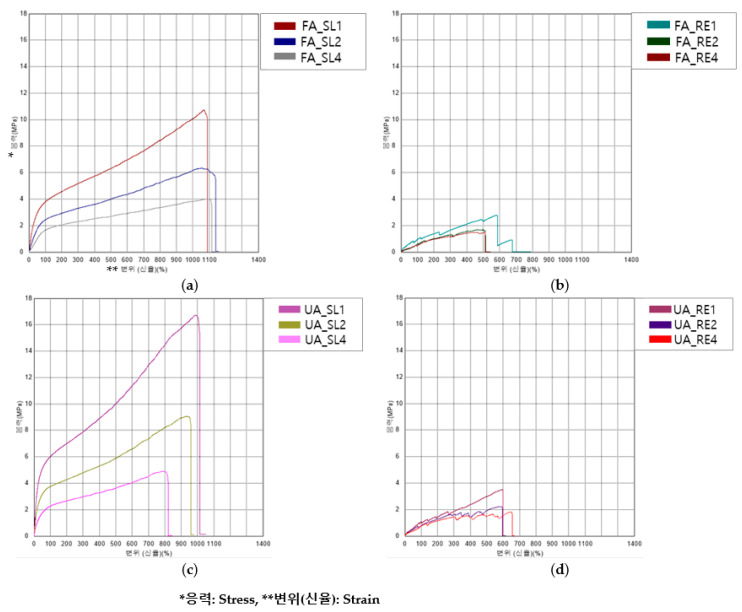
S–S curves of tensile test of 3D-printed forearm and upper arm samples with three different thicknesses: (**a**) FA_SL, (**b**) FA_RE, (**c**) UA_SL, and (**d**) UA_RE.

**Figure 10 polymers-16-00988-f010:**
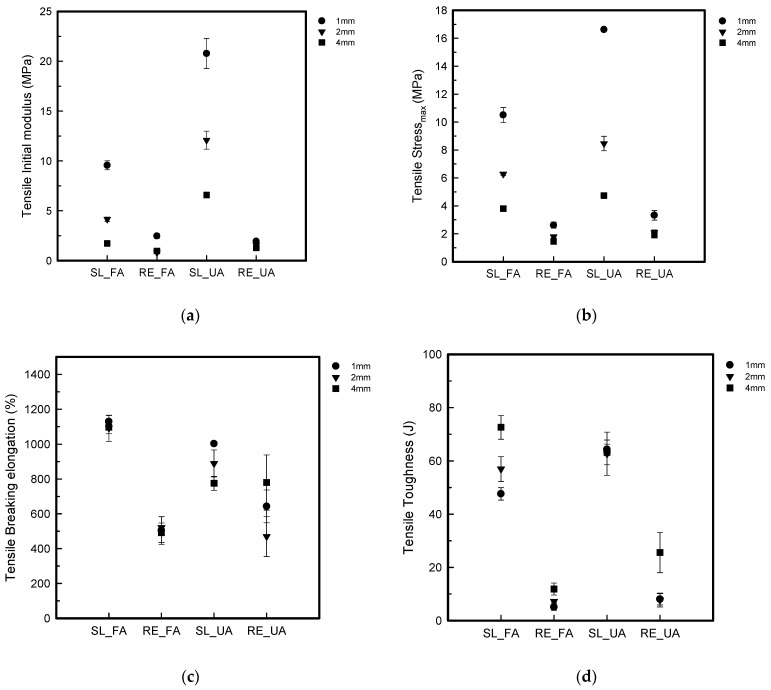
Tensile properties of 3D-printed forearm and upper arm samples with three different thicknesses: (**a**) tensile initial modulus, (**b**) tensile stress_max_, (**c**) tensile strain_max_, and (**d**) tensile toughness.

**Figure 11 polymers-16-00988-f011:**
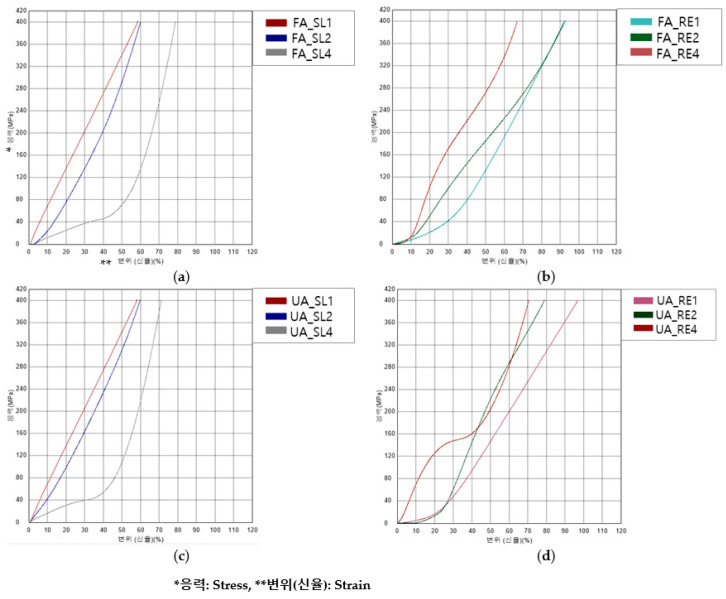
S–S curves of compressive test of 3D-printed forearm and upper arm samples with three different thicknesses: (**a**) FA_SL, (**b**) FA_RE, (**c**) UA_SL, and (**d**) UA_RE.

**Figure 12 polymers-16-00988-f012:**
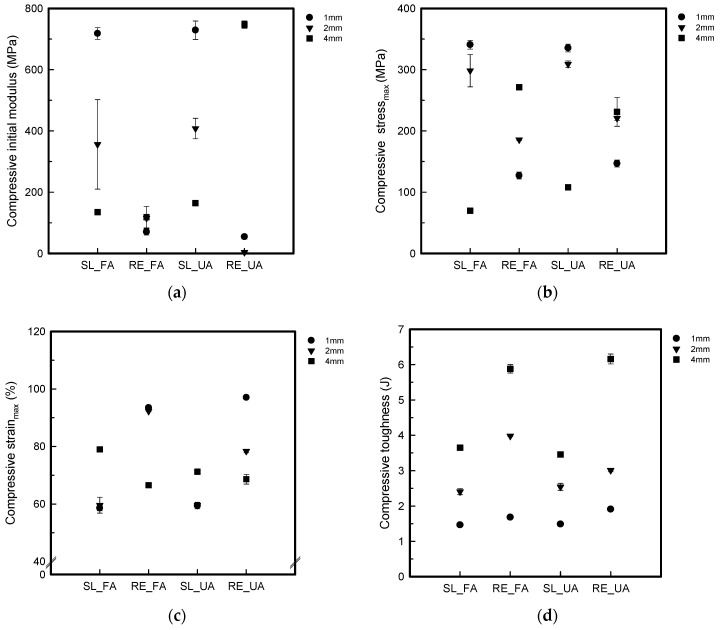
Compressive properties of 3D-printed forearm and upper arm samples with three different thicknesses: (**a**) compressive initial modulus, (**b**) compressive stress_max_, (**c**) compressive strain_max_, (**d**) compressive toughness.

**Table 1 polymers-16-00988-t001:** Sample modeling and size of 3D-printed forearm and upper arm with three different thicknesses.

	Arm Part	PatternImage	Thickness(mm)	Size(mm^3^)	Sample Code
Robot	ModelingImage	Forearm(FA)	Upper Arm (UA)
Forearm (FA)	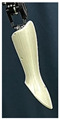	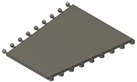	Solid (SL) 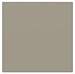	1	165.0 × 130.0 × 1.0	FA_SL1	UA_SL1
2	165.0 × 130.0 × 2.0	FA_SL2	UA_SL2
4	165.0 × 130.0 × 4.0	FA_SL4	UA_SL4
Upper arm (UA)	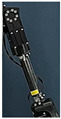	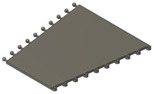	Re-entrant (RE) 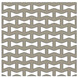	1	190.0 × 155.0 × 1.0	FA_RE1	UA_RE1
2	190.0 × 155.0 × 2.0	FA_RE2	UA_RE2
4	190.0 × 155.0 × 4.0	FA_RE4	UA_RE4

**Table 2 polymers-16-00988-t002:** Sliced images for modeling of 3D-printed forearm and upper arm with three different thicknesses.

Sample Code	Slicing		Sample Code	Slicing	
	Top	Front		Top	Front
FA_SL1	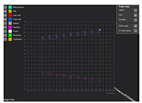	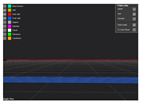	UA_SL1	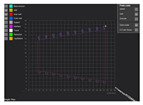	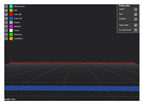
FA_SL2	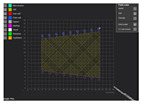	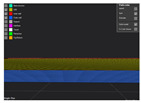	UA _SL2	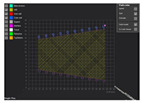	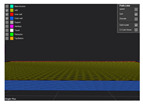
FA_SL4	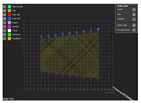	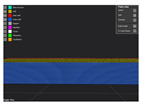	UA _SL4	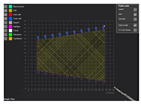	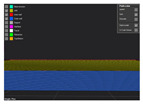
FA_RE1	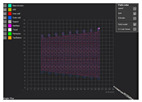	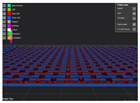	UA _RE1	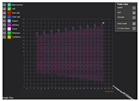	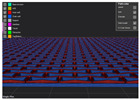
FA_RE2	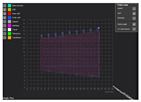	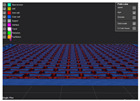	UA _RE2	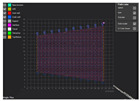	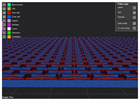
FA_RE4	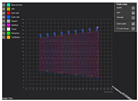	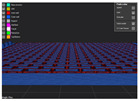	UA _RE4	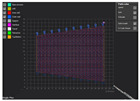	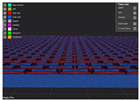

**Table 3 polymers-16-00988-t003:** Sample image of 3D-printed forearm and upper arm with three different thicknesses.

Sample Code	Sample Image	Sample Code	Sample Image	Sample Code	Sample Image	Sample Code	Sample Image
FA_SL1	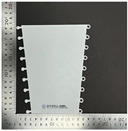	FA_RE1	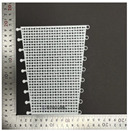	UA_SL1	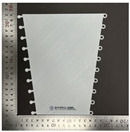	UA_RE1	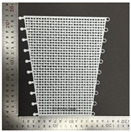
FA_SL2	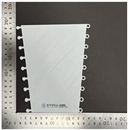	FA_RE2	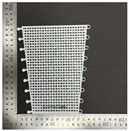	UA_SL2	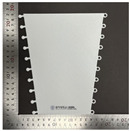	UA_RE2	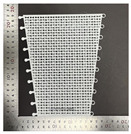
FA_SL4	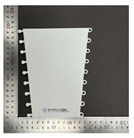	FA_RE4	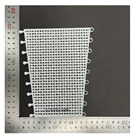	UA_SL4	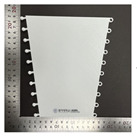	UA_RE4	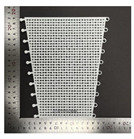

## Data Availability

The data sets used and analyzed during the current study are available from the corresponding author upon reasonable request.

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
