# Peer review of "Characterization of Exterior Parts for 3D-Printed Humanoid Robot Arm with Various Patterns and Thicknesses"

_polymers, 2024, doi:10.3390/polym16070988_

Round 1
Reviewer 1 Report
Comments and Suggestions for Authors
- Abstract should be rewritten, with paying attention to the grammar and sentence flow.
- Sentences in the first paragraph of Introduction are short, and by what is written sound colloquial.
- Introduction should be rewritten in more academic manner.
- Referring to figures and tables should be different.
- Figure 3 - This is not a representation of process. This is figure of universal testing machine with different grasps for different mechanical testing's. Figures with actual specimens would be appropriate. And actual 3D printed specimens should also be presented.
- Stress-Strain curves should be actual curves from the universal testing machine’s software.

Comments on the Quality of English LanguageEnglish must be improved in whole paper, in every manner.
Author Response
We are appreciated for your detailed review and comment on this study.
We have answered about comments carefully.
And the revised parts are marked in red in the attachment of manuscript.
Please confirm of reviewer report in the below and see the attachment.
Thank you.
- Comments and Suggestions for Authors
Q1. Abstract should be rewritten, with paying attention to the grammar and sentence flow.
A1. Yes, based on your opinion we have made changes to the abstract.
< You can find it on the page 1 of the manuscript >
Abstract Currently, metal is the most common exterior material used in robot development due to the need to protect the motor. However, as soft, wearable, and humanoid robots are gradually being developed, many robot parts need to be converted into artificial skin using flexible materials. In this study, in order to develop soft exterior parts for robots, we intend to manufacture exterior robot arm parts via fused filament fabrication (FFF) 3D printing according to various structural and thickness conditions and analyze their mechanical properties. The exterior parts of the robot arms were manufactured utilizing Shore 95 A TPU (eTPU, Esun, China), which is renowned for its softness and exceptional shock absorption characteristics. The exterior robot arm parts were modeled in two parts, the forearm and upper arm, by applying solid (SL) and re-entrant (RE) structures and thicknesses of 1, 2, and 4 mm. The mechanical properties were analyzed through the use of three-point bending, tensile, and compression testing. All of the characterizations were analyzed using a universal testing machine (AGS-X, SHIMADZU, Japan). After testing the samples, it was confirmed that the RE structure was easily bendable towards the bending curve and required less stress. In terms of the tensile tests, the results were similar to the bending tests; to achieve the maximum point, less stress was required, and for the compression tests, the RE structure was able to withstand the load compared to the SL structure. Therefore, after analyzing all three thicknesses, it was confirmed that the RE structure with a 2 mm thickness had excellent characteristics in terms of bending, tensile, and compressive properties. Therefore, the re-entrant pattern with a 2 mm thickness is more suitable for manufacturing a 3D-printed humanoid robot arm.
Q2. Sentences in the first paragraph of Introduction are short, and by what is written sound colloquial.
A2. Yes, thank you for your comment. We have revised the introduction and used more long sentences based on your comments.
< You can find it on the page 1 & 2 of the manuscript >
Q3. Introduction should be rewritten in more academic manner.
A3. Yes, based on your opinion we have rewritten the introduction in more academic manner.
< You can find it on the page 1 & 2 of the manuscript >
Q4. Referring to figures and tables should be different.
A4. Figures and tables have been changed to journal format accordingly.
Q5. Figure 3 - This is not a representation of process. This is figure of universal testing machine with different grasps for different mechanical testing’s. Figures with actual specimens would be appropriate. And actual 3D printed specimens should also be presented.
A5. Yes, thank you for your comment. According to your guidance we have changed the figures to the figures with actual specimen in it for better understanding.
< You can find it on the page 5 & 6 of the manuscript >
Q6. Stress-Strain curves should be actual curves from the universal testing machine’s software.
A6. Thank you for your comment, but the graphs created in the universal testing machine’s software is too small to understand in details. Original graphs have been attached below for your reference but the graphs have not been changed in the manuscript.
< You can find it on the page 10, 13 & 16 of the manuscript >
Figure . Bending test graph for forearm for both solid and re-entrant pattern
Figure . Bending test graph for upper arm for both solid and re-entrant pattern
Figure . Tensile test graph for forearm for both solid and re-entrant pattern
Figure . Tensile test graph for upper arm for both solid and re-entrant pattern
Figure . Compressive test graph for forearm for both solid and re-entrant pattern
Figure . Compressive test graph for upper arm for both solid and re-entrant pattern
- Areas for improvement of Pdf: According to your comments the improvement from the PDF has been done and below are the modifications that were done for your reference.
Q1. ?
A1. Materials is changed to parts.
< You can find it on the page 1 & 18 of the manuscript >
Q2. Please, rewrite this sentence.
A2. The sentence has been rewritten
"The exterior parts of the robot arms were manufactured utilizing Shore 95 A TPU (eTPU, Esun, China), renowned for its softness and exceptional shock absorption characteristics."
< You can find it on the page 1 of the manuscript >
Q3. Materials or parts?
A3. Materials has been changed to parts
< You can find it on the page 1 of the manuscript >
Q4. Not needed to be placed in the abstract.
A4. The sentence has been removed as per the guidance.
< You can find it on the page 1 of the manuscript >
Q5. Mechanical properties were analyzed through.... testings.
A5. The sentence has been changed to "The mechanical properties were analyzed through three-point bending, tensile and compression testing."
< You can find it on the page 1 of the manuscript >
Q6. Please, rewrite with caution whole paragraph.
A6. We have revised the introduction and used more long sentences.
"Traditionally, robot exteriors were predominantly composed of rigid materials, limiting their physical interaction capabilities with humans. These exterior materials were designed to encase various components such as motors, actuators, sensors, and associated electronic wiring to safeguard the internal mechanisms. Typically, metals or plastics were employed in the manufacturing process to ensure robust protection for the robot. However, with the advancement of humanoid robotics, there has been a growing recognition of the necessity for softer and more flexible exteriors to facilitate improved interaction with humans. Achieving a soft robot exterior can be realized through careful selection of materials or by designing the structure accordingly. An emerging and promising approach to crafting soft robot exteriors involves the utilization of auxetic structures in meta-materials. Auxetic structures exhibit a unique property wherein they can expand laterally in one or more perpendicular directions when subjected to axial extension. This distinctive behavior grants them desirable mechanical properties such as elasticity, shock absorption, and strain resilience. By incorporating auxetic meta-materials into the exterior composition, robots can benefit from enhanced protection against external impacts while still maintaining the required flexibility for human interaction. This approach holds significant promise in advancing the field of robotics by enabling robots to operate safely and effectively in close proximity to humans."
< You can find it on the page 1 & 2 of the manuscript >
Q7. Rewrite text.
A7. The text has been rewritten to "they are typically composed of softer, more flexible, and absorbent materials".
< You can find it on the page 2 of the manuscript >
Q8. Rewrite text.
A8. The text has been changed to " This technique enables precise control over both heating and nozzle size, allowing adjustments to be made according to specific requirements"
< You can find it on the page 2 of the manuscript >
Q9. This part should be reorganized, stating first the material, then 3D printer, with the printing parameters.
A9. This part is reorganized to “TPU filament (eTPU-95A, Esun Industrial Co. Ltd., China) with a diameter of 1.75 mm, density of 1.43 g/cm³, and a Shore hardness of 95A in white color, was utilized. Printing was conducted using a fused filament fabrication (FFF) technique with a 3D printer (Cubicon single plus, Cubicon Co. Ltd., Korea) equipped with a 0.4 mm diameter nozzle. The 3D model of the humanoid robot arm was saved in *.stl files. Slicing and printing were performed using Cubicreator4 V4.4.0 slicing program (Cubicon Co. Ltd., Korea) under the following conditions: nozzle temperature set at 225 ℃, bed temperature at 65 ℃, printing speed of 60 mm/sec, and infill pattern and density set to zigzag and 20%, respectively. Transformation of *.stl files into printable *.g-code files was executed within the slicing program."
< You can find it on the page 3 of the manuscript >
Q10. This figure is not needed, because value of the diameter is already written.
A10. Figure has been removed as per guidance.
< You can find it on the page 3 of the manuscript >
Q11. This part should be connected with previous mentioning of 3D printing device.
A11. This part has been added with the previous mentioning of 3D printing device as per your guidance.
< You can find it on the page 3 of the manuscript >
Q12. Figures with actual specimens would be appropriate.
A12. Figures with actual specimen has been added.
< You can find it on the page 5 & 6 of the manuscript >
Q13. This is a slicing image of model, not actual 3D printed arm. It should be changed in text.
A13. Text has been changed to "Analysis of sliced image for modeling of 3D printed humanoid robot arm with various patterns and thicknesses"
< You can find it on the page 6 of the manuscript >
Q14. Printing times are not part of the characterization.
A14. Yes, thank you for your comment. Although printing time is not a part of characterization, but it explains the difference between the time taken, weight and thicknesses of two different patterns with various thicknesses so it was added in the characterization.
< You can find it on the page 7 of the manuscript >
- Comments on the Quality of English Language
A: The proof reading for quality of English language has been done and the certificate is attached below for your reference.

Reviewer 2 Report
Comments and Suggestions for Authors
The manuscript by these Authors deals with the preparation (by additive manufacturing) and characterization of the exterior material of the robot arm. The topic is interesting but, in my opinion, the approaches could be improved. Apart from some minor questions highlighted in the attached pdf, my major concern is related to the term "physical properties" used by the Authors. Both in the abstract and introduction they aim at analyzing physical properties but, really, they studied mechanical not physical ones.

Author Response
Reviewer 2
We are appreciated for your detailed review and comment on this study.
We have answered about comments carefully.
And the revised parts are marked in red in the attachment of manuscript.
Please confirm of reviewer report in the below and see the attachment.
Thank you.
- Comments and Suggestions for Authors
The manuscript by these Authors deals with the preparation (by additive manufacturing) and characterization of the exterior material of the robot arm. The topic is interesting but, in my opinion, the approaches could be improved. Apart from some minor questions highlighted in the attached pdf, my major concern is related to the term "physical properties" used by the Authors. Both in the abstract and introduction they aim at analyzing physical properties but, really, they studied mechanical not physical ones.
Areas for Improvement:
A1. Yes, thank you for your comment. According to your guidance the term “physical properties” are changed to mechanical properties in the abstract as well as the context.
< You can find it on the page 1 of the manuscript >
“Abstract: …..In this study, in order to develop soft exterior parts for robots, we intend to manufacture exterior robot arm parts via fused filament fabrication (FFF) 3D printing according to various structural and thickness conditions and analyze their mechanical properties. The exterior parts of the robot arms were manufactured utilizing Shore 95 A TPU (eTPU, Esun, China), which is renowned for its softness and exceptional shock absorption characteristics. The exterior robot arm parts were modeled in two parts, the forearm and upper arm, by applying solid (SL) and re-entrant (RE) structures and thicknesses of 1, 2, and 4 mm. The mechanical properties were analyzed through the use of three-point bending, tensile, and compression testing. All of the characterizations were analyzed using a universal testing machine (AGS-X, SHIMADZU, Japan)…”
< You can find it on the page 1 & 3 of the manuscript >
“Introduction: …..RE structure has more auxetic properties with smaller angle [14]. 3D structure hexagonal RE has auxetic behaviors in multiple directions, and it was also found that mechanical properties of a RE can also be controlled by the RE angle.
… To confirm the suitable conditions for the robot arm’s exterior material, two different patterns and three different thicknesses were applied to the 3DP sample. The mechanical properties were analyzed through bending, tensile, and compressive tests. Based on the research results, we intend to use this material for covering the arms of the humanoid robot.”
< You can find it on the page 18 of the manuscript >
“Conclusion: In this study, we aimed to develop a soft exterior material for humanoid robot arms by fused filament fabrication (FFF) 3D printing. Using Shore 95 A TPU filament, the soft exteriors applied two different patterns & three different thickness were printed and analyzed by mechanical properties….
…. According to the study, using a 2 mm thick RE pattern is ideal for making the 3DP humanoid robot arm. The RE structure's great bending, tensile, and compressive qualities make it a perfect choice for ensuring the strength and performance of the robot arm, highlighting its excellent mechanical properties and suitability for the intended application.”
- Areas for improvement of Pdf: According to your comments the improvement from the PDF has been done and below are the modifications that were done for your reference.
Q1. remove line
A1. It has been changed to "However, as soft, wearable, and humanoid robots are gradually being developed, many parts of the robot need to be converted into artificial skin using flexible materials."
< You can find it on the page 1 of the manuscript >
Q2. Please indicate in the abstract just the acronym, for other material specifics report in experimental section.
A2. Yes, as per you it has been changed and added to the experimental section.
Q3. and
A3. It has been changed to "Auxetic materials are structures that have a negative Poisson ratio [6] and constructed in such a way that has gap in between the structure of the sample so it shows the properties of permeability, absorption, indentation resistance, shear resistance, toughness and easy bending resulting in synclasticity."
< You can find it on the page 2 of the manuscript >
Q4. carried out
A4. It has been changed to "Previously a study was carried out on the motion control of a 3D printed(3DP) fingers with two types of sample, cap and RE samples."
< You can find it on the page 2 of the manuscript >
Q5. follow the journal format
A5. Yes, thank you for your guidance, it has been modified to journal format.
Q6. mechanical
A6. Physical properties have been changed to mechanical properties.
< You can find it on the page 1, 3 &18 of the manuscript >
Q7. in order to guarantee these aspects the test the authors carried out are not sufficient
A7. The sentence has been changed to "According to the study, using a 2 mm thick RE pattern is ideal for making the 3DP humanoid robot arm. The RE structure's great bending, tensile, and compressive qualities make it a perfect choice for ensuring the strength and performance of the robot arm, highlighting its excellent mechanical properties and suitability for the intended application" for better understanding.”
< You can find it on the page 18 of the manuscript >
- Comments on the Quality of English Language
A: The proof reading for quality of English language has been done and the certificate is attached below for your reference.

Round 2
Reviewer 1 Report
Comments and Suggestions for Authors
Please, add the actual stress-strain curves from the machines software, as it is the only acceptable representation. Implement these curves which you have sent in the answer for tensile and compression tests, and bending ones to be the same, all with gridlines. The graphs you have chosen with "screaming" lines are not appropriate, because lines presented like this need to be averaged in order to read the values.
For compressive diagrams, please add the rest of the lines in diagrams.
Engineering measurements for stress-strain diagrams should be presented in Pa (i.e. MPa), it can't be only N, as it does not mean anything without /m2.
Also, are these engineering stress-strain diagrams? Or actual? It should be mentioned.
Author Response
Thank you for your comment on review.
We have answered about comments carefully. The changed parts are marked in red.
As you commented, we modified the S-S graph of bending/tensile/compressive properties. Data was extracted from the program and confirmed by changing the unit to MPa. Thank you for commenting on the fact that we did not consider thickness in advance. Accordingly, all changed tendency and data have been revised in the contents.
Also, in the case of the screaming you mentioned, it is especially confirmed in the 1mm sample of bending property. And it is thought to be a phenomenon that appears due to the thinness of the sample when measuring. All measured data values were extracted and analyzed using the values confirmed in the program.
We have attached the revised manuscript.
Once again, Thank you for your comments. Please confirm.
Best regards.

Reviewer 2 Report
Comments and Suggestions for Authors
The manuscript can be accepted for publication.
Author Response
Thank you for your approval.
As the unit of data changed through other reviews, the graph and data in the text were modified.
We marked the changed part in red.
We have attached the revised manuscript, so please check it.
Please check it, and thank you.
